# Do pupillary responses during authentic slot machine use reflect arousal or screen luminance fluctuations? A proof-of-concept study

Andy J. Kim[1,☯,¤a], W. Spencer Murch[1,☯,¤b], Eve H. Limbrick-Oldfield[1], Mario A. Ferrari[1], Kent I. MacDonald[1], Jolande Fooken[2,3], Mariya V. Cherkasova[4,5], Miriam Spering[2,4], Luke Clark[1,4]*

1 Centre for Gambling Research at UBC, Department of Psychology, University of British Columbia, Vancouver, British Columbia, Canada, 2 Department of Ophthalmology and Visual Sciences, University of British Columbia, Vancouver, British Columbia, Canada, 3 Centre for Neuroscience Studies, Queen's University, Kingston, Ontario, Canada, 4 Djavad Mowafaghian Centre for Brain Health, University of British Columbia, Vancouver, British Columbia, Canada, 5 Department of Psychology, West Virginia University, Morgantown, West Virginia, United States of America

☯ These authors contributed equally to this work.
¤a Current address: Department of Psychology and Neuroscience, Dalhousie University, Halifax, Nova Scotia, Canada
¤b Current address: Department of Sociology and Anthropology, Concordia University, Montréal, Quebec, Canada
* luke.clark@psych.ubc.ca

**Data Availability Statement:** For transparency and reproducibility, our data and analysis scripts have

## Abstract

Modern slot machines are among the more harmful forms of gambling. Psychophysiological measures may provide a window into mental processes that underpin these harms. Here we investigated pupil dilation derived from eye tracking as a means of capturing changes in sympathetic nervous system arousal following outcomes on a real slot machine. We hypothesized that positively reinforcing slot machine outcomes would be associated with increases in arousal, reflected in larger pupil diameter. We further examined the contribution of game luminance fluctuations on pupil diameter. In Experiment 1A, experienced slot machine gamblers ($N = 53$) played a commercially-available slot machine in a laboratory for 20 minutes while wearing mobile eye tracking glasses. Analyses differentiated loss outcomes, wins, losses-disguised-as-wins, and (free-spin) bonus features. Bonus features were associated with rapid increases in pupil diameter following the onset of outcome-related audiovisual feedback, relative to losses. In Experiment 1B, luminance data were extracted from captured screen videos (derived from Experiment 1A) to characterize on-screen luminance changes that could modulate pupil diameter. Bonus features and wins were associated with pronounced and complex fluctuations in screen luminance ($\approx 50$ L and $\approx 25$L, respectively). However, the pupil dilation that was observed to bonus features in Experiment 1A coincided temporally with only negligible changes in screen luminance, providing partial evidence that the pupil dilation to bonus features may be due to arousal. In Experiment 2, 12 participants viewed pairs of stimuli (scrambled slot machine images) at luminance difference thresholds of $\approx 25$L, $\approx 50$L, and $\approx 100$L. Scrambled images presented at luminance differences of $\approx 25$L

been archived at https://doi.org/10.5683/SP3/MQVSP3.

**Funding:** This study was funded by a Discovery Award to LC from the Natural Sciences and Engineering Research Council of Canada (RGPIN-2017-04069)(https://www.nserc-crsng.gc.ca/index_eng.asp), and the core funding of the Centre for Gambling Research at UBC, which is supported by the Province of British Columbia government and the British Columbia Lottery Corporation (BCLC; a Canadian Crown Corporation)(https://www.bclc.com). The Centre for Gambling Research at UBC provided salary support to AJK, KM, EHLO, and LC. AJK also received an Undergraduate Student Research Award from the Natural Sciences and Engineering Research Council of Canada, and is now supported by a Nova Scotia Graduate Scholarship – Masters, and Maritime Spor Support Unit Award - Masters. WSM and MAF held graduate fellowships from the Natural Sciences and Engineering Research Council of Canada (NSERC). WSM now holds a Horizon Postdoctoral Fellowship funded by Concordia University, the Concordia Office of the Vice President of Research and Graduate Studies, the Concordia Research Chair on Gambling Studies, Fonds de Recherche du Québec - Fonds Société et Culture, and the Mise sur toi Foundation. JF holds a Deutsche Forschungsgemeinschaft (DFG) Research Fellowships (grant FO 1347/1-1). MS holds an NSERC Discovery Grant and NSERC Accelerator Supplement (RGPIN 418493), and a Canada Foundation for Innovation grant that funded the mobile eye-tracking device. The funders had no role in study design, data collection and analysis, decision to publish, or preparation of the manuscript.

**Competing interests:** I have read the journal's policy and the authors of this manuscript have the following competing interests: The Centre for Gambling Research at UBC receives funding from the Province of British Columbia and the British Columbia Lottery Corporation (BCLC), a Canadian Crown Corporation. The slot machines used in the present study were provided to the Centre for Gambling Research by the BCLC. EHLO has received a speaker honorarium from the Massachusetts Council on Compulsive Gambling (USA). She has accepted travel or accommodation for speaking engagements from the National Council for Responsible Gambling (USA), the International Multidisciplinary Symposium on Gambling Addiction (Switzerland), and the Responsible Gambling Council (Canada). She has not received any further direct or indirect payments from the gambling industry or groups substantially

and greater were sufficient to cause pupillary responses. Overall, pupillometry may detect event-related changes in sympathetic nervous system arousal following gambling outcomes, but researchers must pay careful attention to substantial in-game luminance changes that may confound arousal-based interpretations.

## Introduction

Eye tracking technology is becoming more widely used in gambling research [1–4], and equipment costs have dropped substantially in recent years. In addition to measuring overt visual attention, these developments have enabled psychophysiological measures involving the eye's pupil. Phasic changes in pupil diameter can indicate psychological reactions to internal or external events [5–7]. This study explores the acquisition and interpretation of pupillary data obtained during real-world gambling, in the specific context of authentic slot machine games.

The recording of psychophysiological measures during gambling may provide a window into the psychological processes involved in gambling (e.g., positive reinforcement, excitement) and may further uncover individual differences relevant to the risk of experiencing gambling-related harms. Beginning in the 1980s (e.g., [8]), a range of physiological measures have been examined in relation to gambling, including various cardiographic indices [9–12], electrodermal indices [13–16], muscular indices [17–19], and body temperature [20]. Some of these measures have been further linked to problem gambling status, although not consistently so [21–23]. From a methodological perspective, physiological measures vary in several respects, including the speed of response (enabling the detection of 'event-related' responses to gambling wins or losses) and the specificity of any physiological interpretation (e.g., to sympathetic vs parasympathetic nervous system activity; [24, 25]).

The present study focuses on physiological signals during slot machine gambling, one of the more harmful forms of gambling [26–28]. Modern slot machines are highly engineered games, and the harms associated with their use have been linked to a number of 'structural characteristics' including fast game speed [29], regular reinforcement [30], and intense audio-visual feedback [31, 32]. Traditional perspectives on gambling motives and the nature of gambling reinforcement have emphasized the role of excitement (i.e., thrill or 'rush') [33–35]. But recent work has also linked the problematic use of slot machines to a state of immersion, aligned with motives of coping and escape [36, 37], and notably, excitement and immersion could be hypothesized to have quite distinct psychophysiological signatures (e.g., [12]).

To date, psychophysiological studies on slot machine gambling have primarily examined 'tonic' changes, occurring over several minutes of play [22, 38–40]—often with ambiguous results [41]. In terms of event-related (or 'phasic') analyses, changes in skin conductance have been described following wins and near-miss outcomes, compared to losses [13, 42, 43], but these studies have used rather simplified slot machine 'simulations'. As skin conductance levels peak around 4—5s post-stimulus, this physiological measure is still limited in its suitability for use with modern, authentic slot machines, on which successive bets can often be placed every 3–4 seconds [44]. Thus, alternative psychophysiological measures that display a faster response profile could have considerable value in characterizing gambling arousal changes in relation to specific in-game events.

The pupil dilator muscle reacts within 200–2,500 milliseconds of a stimulus [45], a property that is well suited to 'event related' assessments. Pupillary responses in consistently-lit environments indicate activation of the sympathetic nervous system [46–49], and are further modulated by noradrenergic [50] and dopaminergic [51] neurotransmission. Pupillary measures are

funded by gambling. MAF has received a speaker honorarium from the British Columbia Lottery Corporation (BCLC). MVC has received funding from the International Center for Responsible Gaming. She has received a speaker honorarium and travel and accommodation from the Responsible Gaming Association of New Mexico (USA). She has accepted a travel award from the International Multidisciplinary Symposium on Gambling Addiction (Switzerland). LC is the Director of the Centre for Gambling Research at UBC. LC has received speaker/travel honoraria from the National Association of Gambling Studies (Australia) and the International Center for Responsible Gaming (USA). He has received academic consulting fees from Gambling Research Exchange Ontario (Canada), GambleAware (UK), and the International Center for Responsible Gaming (USA). He has not received any further direct or indirect payments from the gambling industry or groups substantially funded by gambling. He has received royalties from Cambridge Cognition Ltd. relating to neurocognitive testing. AJK, WSM, KIM, JF and MS report no conflicts of interest. This does not alter our adherence to PLOS ONE policies on sharing data and materials.

increasingly studied within research on decision-making, showing sensitivity to key components of gambling including choice, uncertainty, reward anticipation, and reward receipt [52–57]. For example, in a study using a two-choice lottery task [31], changes in pupil diameter during the choice and anticipation intervals were associated with both the size of the available reward and its probability, and pupil dilation was further increased by the addition of audiovisual feedback to the task.

The overarching aim of the present study was to examine pupillary changes in the context of modern, authentic slot machine gambling. Experiment 1A analyzed pupil diameter from an eye tracking experiment [58] in which experienced slot machine gamblers played a genuine slot machine game for 20 minutes. We distinguished four types of gambling outcomes for assessing event-related changes in pupil diameter. In addition to standard 'wins' (where the payout exceeds the bet) and 'losses' (i.e., zero payout outcomes), we also coded losses-disguised-as-wins (LDWs) [59] as celebrated outcomes where the amount won is less than the bet; and free-spin bonus features [60, 61], as rare events that are accompanied by distinctive and intense audiovisual cues, and are known to be highly appealing outcomes to regular slot machine gamblers [62]. We hypothesized that these three types of reinforcing outcomes would be associated with significant pupil dilation (i.e., increased pupil diameter) relative to losses.

## Experiment 1A

### Methods

**Participants and procedure.** This paper reports pupillary analyses of an eye tracking experiment reported in [58], involving 53 community-recruited slot machine gamblers (32 men and 21 women; mean age = 33.53, SD = 12.30). Participants were recruited through craigslist.ca advertisements. They were at least 19 years old and reported playing a slot machine in the past 12 months. All participants scored < 8 (for high risk of gambling problems) on the Problem Gambling Severity Index; [63]). They had no history of neuropsychiatric or ophthalmic disease, psychotropic medication use, or recent/severe traumatic brain injury. They had normal or corrected-to-normal visual acuity with prescription strength between -4 and +4 diopters.

Participants were paid $20 CAD for their participation and received any earnings on the slot machine task as a bonus (up to $20). All experimental protocols were approved by UBC's Behavioural Research Ethics Board, and participants provided written informed consent.

*Apparatus*. Participants gambled using an endowment of $40 cash on a genuine slot machine ("Buffalo Spirit," Scientific Games Co., Las Vegas, NV) for 20 minutes, or until they ran out of credit (see S1 Text for additional procedure details). Participants initiated each spin (akin to a 'trial' in a psychology experiment) by pressing the bet button on the right-hand side of the slot machine fascia. The reels spun for up to 6 seconds [44] and stopped to reveal one of four possible outcomes: a loss, win, loss-disguised-as-a-win, or a bonus round. Following each spin, any payout and associated audiovisual feedback was presented. Losses have no positively-reinforcing payouts or audiovisuals, and are the most common outcome. Wins and losses-disguised-as-wins involve both a payout and audiovisual feedback, but on losses-disguised-as-wins, the amount won is less than the amount wagered [59, 64]. During audiovisual feedback for wins and losses-disguised-as-wins, the value displayed on the slot machine's credit and win windows incremented over several seconds, and therefore the duration of the feedback interval to these outcomes is variable and related to the size of the win. Furthermore, the exact value of a given win, loss-disguised-as-a-win, or bonus feature was only known to participants *at the end* of any outcome-related audiovisual feedback. After the audiovisuals finished, the device

became idle, and waited for the participant to initiate the next spin. We defined this epoch as the inter-trial interval.

The free spin bonus feature was awarded by three special (and over-sized) 'bonus' symbols visible on the screen by the end of the spin (see Fig 3). This triggered 15 or more free spins that proceeded automatically (i.e., without the player being required to press the bet button). The free spin bonuses doubled the value of any wins and had a unique music track for the duration of the feature. On a bonus feature, any winnings accrued were transferred to the participant's credit balance only after the last free spin had ended. Because of their uniqueness and lack of participant intervention, we treated the multiple spins within a bonus feature as a single entity.

During the slot machine gambling session, participants wore a pair of mobile eye tracking glasses that provided real-time, natural gaze behavior from both eyes at a rate of 60 Hz (SMI, Teltow, Germany). Eye position data were calibrated using three corner symbols on the game screen and recorded to a Samsung Galaxy Note 4 affixed to the back of the participant's seat. Participants were able to sit at a comfortable distance from the game; data could be extracted from the eye tracking glasses without requiring participants' heads to be stabilized.

**Data analysis.** Data were pre-processed using SMI's proprietary analysis software (BeGaze 3.7). A binocular average pupil diameter in millimeters (mm) was extracted at 60 Hz for the duration of the slot machine session. To link the pupil data with slot machine outcomes, we derived a time series of game events from a video capture of the slot machine screen, using an internal video splitter and image recognition software that we developed (see S1 Text).

To calculate pupillary responses, we defined three epochs on each spin (see Fig 1): 1) a Baseline epoch was calculated as the mean pupil diameter during the last 200 ms of the reel spin, 2) an audiovisual feedback epoch from 200 ms to 2,500 ms [45] after the reels stopped (T1; this epoch captures the response to the initial audiovisual feedback), and 3) an epoch from 200 ms to 2,500 ms after the *offset* of the audiovisual feedback (T2; this epoch captures the response to the final amount won). We were interested in both the start (T1) and end (T2) of audiovisual feedback because T1 indexed the first instance when gamblers became aware that reinforcement would be delivered, and T2 indexed their response to finding out the actual monetary value of the reinforcement. If the duration of the feedback was under 2.5s, there was some degree of overlap in the T1 and T2 epochs. This overlap varied by outcome type, with LDWs having the shortest average feedback duration and therefore the most overlap. Loss outcomes have no associated audiovisual feedback, and thus the T2 offset epoch cannot be specified.

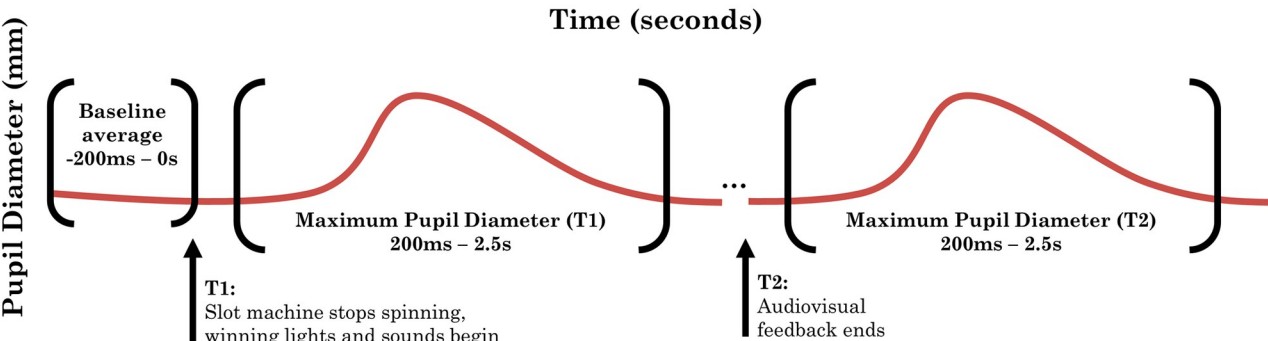

**Fig 1. Recording epochs for pupillary responses.** Note that on Loss outcomes, there is no ensuing audiovisual feedback and therefore T2 cannot be modelled separately from T1. On the other outcomes, if the duration of audiovisual feedback was less than 2.5 seconds, the T1 and T2 epochs would overlap; see S1 Table for average durations.

In order to measure changes in pupil diameter, we calculated the peak pupil dilation values within the T1 and T2 epochs and calculated the percentage change from the average diameter recorded during the Baseline epoch for that trial [45]. Missing pupil diameter data due to blinks and saccades were estimated using linear interpolation. If more than 18% of samples in a given epoch were missing due to blinks or saccades, it was discarded, as per published recommendations [45]. We excluded 43 outlying values that were more than three standard deviations from the mean response, likely due to equipment error. In total, 5,934 values were included. The resulting distribution of data appeared normal.

A fixed-effects regression model was created to predict pupil diameter for the three positively-reinforced outcome types relative to losses [44, 65, 66]. We fitted three dummy-coded outcome types (wins, losses-disguised-as-wins, and bonus features), using losses as the reference category. We also included predictors representing trial number and credit balance (i.e., credits held at the start of the current spin). Trial number was included to examine changes in pupil magnitude over the duration of the task. The 'credit balance' variable was included to account for any variance that the broader financial context (the participant's 'performance') could have on their responses to discrete outcomes.

For transparency and reproducibility, our data and analysis scripts have been archived at https://doi.org/10.5683/SP3/MQVSP3. Please refer to the S1 Text for additional details.

## Results

Descriptive data on the number of trials per outcome type are presented in the S1 Text and S1 Table. Fig 2A shows changes of pupil dilation as a function of different time epochs and audiovisual events for a model participant. Although pupil diameter was relatively invariant for losses and losses-disguised-as-wins, we observed event-related changes in pupil diameter for wins and free spin bonus features. Inferential analyses showed that, relative to loss trials, free spin bonus features were associated with a significant pupillary response in the T1 epoch following the onset of audiovisual feedback ($B = 3.18$, $t(5830) = 3.83$, $p < .001$, Table 1, Fig 2C). Wins ($B = 0.09$, $t(5830)$ 0.43, $p = .668$) and losses-disguised-as-wins ($B = -0.20$, $t(5830) = -0.96$, $p = .338$) did not differ from losses at T1. Following the offset of audiovisual feedback (T2 epoch), we saw a slight decrease in pupil diameter for wins relative to losses ($B = -1.09$, $t(5830) = -4.99$, $p < .001$), and a larger decrease for bonus features ($B = -3.24$, $t(5830) = -3.97$, $p < .001$). There was no significant effect at T2 for losses-disguised-as-wins ($B = -0.25$, $t(5830) = -1.22$, $p = .223$). Trial number was negatively related to pupillary response magnitude ($B > -0.01$, $t(5830) = -3.32$, $p = .001$, Table 1), indicating that event-related increases in pupil diameter diminished over the course of the session. Participants' credit balance was not a significant predictor of pupil size ($B > -0.01$, $t(5830) = -1.71$, $p = .088$). Overall, the model accounted for 55% of the variance in task-related pupillary responses ($R^2_{adj} = .55$).

## Experiment 1A discussion

We investigated pupillary responses as a potential physiological marker of reward processing among experienced gamblers using an authentic slot machine. We created a regression model that accounted for 55% of the variance in pupillary responses. Pupil diameter increased by approximately 3.2% in response to the onset of audiovisual feedback on free-spin bonus features, relative to loss trials, consistent with sympathetic nervous system activation. For comparison against another appetitive task, similar effects were observed in healthy participants consuming chocolate [45]. We detected no significant increases in pupil diameter to the equivalent epoch following wins and losses-disguised-as-wins; the pupillary response on these outcomes was generally flat (Fig 2B), and their associated confidence intervals considerably

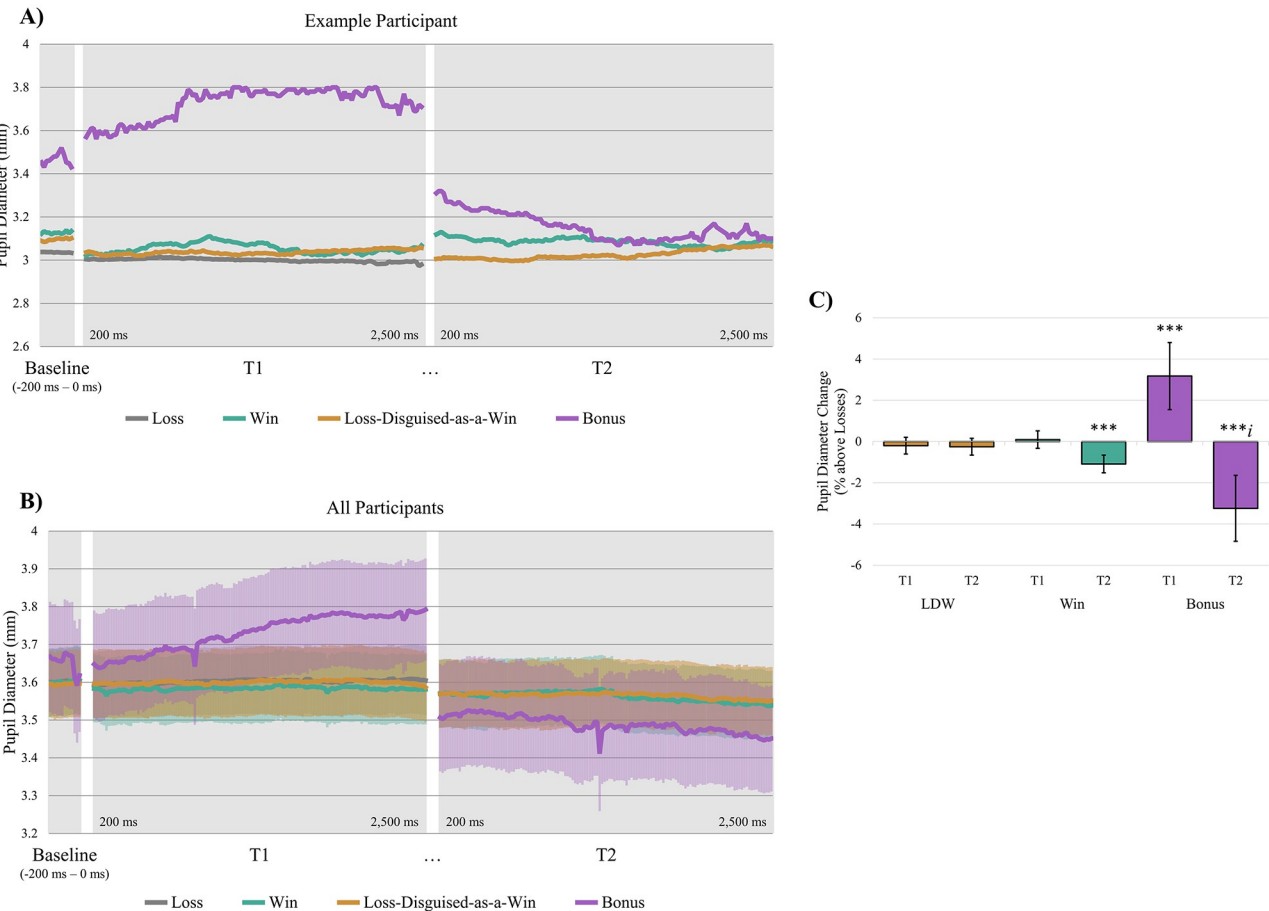

**Fig 2. Change in pupil diameter by outcome type in a representative participant and the overall group.** (A) Pupil diameter trace for a representative participant through Baseline, and at the start (T1), and end (T2) of audiovisual feedback. Lines represent the different outcome types. Loss trials have no T2. (B) Mean pupil diameter for all participants. Lines represent the mean of different outcome types. Shaded areas represent the standard error between subjects for that point in time. (C) Predicted pupillary response to different slot machine outcomes, as percentage change from Baseline, minus the response for loss outcomes. Asterisks represent the effect of the outcome relative to the reference category, loss trials. Bars represent the 95% confidence interval. LDW = Loss-Disguised-as-a-Win, * $p < .05$, ** $p < .01$, *** $p < .001$.

overlapped the pupillary activity following losses (Fig 2C). We also observed unexpected decreases in pupil diameter (i.e., pupil contraction) following the offset of audiovisual feedback (the T2 epoch) to bonus features and wins (see S1 Text for further discussion).

## Experiment 1B

Game-related changes in screen luminance could also drive changes in pupil diameter, given the established effects of screen brightness on the pupillary light reflex [7]. In visual psychophysics, researchers pay careful attention to stimulus luminance in designing their tasks, often keeping luminance constant to avoid confounding the effect of interest with changes in luminance. Naturally, there is no such reason for slot machine designers to intentionally keep luminance fixed. If bonus features, for example, are accompanied by significant decreases in screen luminance, this may indicate that the pupil dilation observed in Experiment 1A is not mediated by physiological arousal and/or excitement but instead driven by luminance changes. By the same reasoning, any systematic luminance-related increases on wins and LDWs could

**Table 1. Event-related change in pupil diameter as a percentage of baseline by trial number, prior credits, outcome type, and outcome phase.**

| Factor | B | 95% CI | t(5830) | p |
|---|---|---|---|---|
| Trial Number$_{(1)}$ | >-0.01 | [-0.01, >-0.01] | -3.32 | .001*** |
| Credit Balance$_{(4,000)}$ | >-0.01 | [>-0.01, <0.01] | -1.71 | .088 |
| Win T1$_{(Loss)}$ | 0.09 | [-0.33, 0.52] | 0.43 | .668 |
| T2$_{(Loss)}$ | -1.09 | [-1.51, -0.66] | -4.99 | < .001*** |
| LDW T1$_{(Loss)}$ | -0.20 | [-0.60, 0.21] | -0.96 | .338 |
| T2$_{(Loss)}$ | -0.25 | [-0.66, 0.15] | -1.22 | .223 |
| Bonus T1$_{(Loss)}$ | 3.18 | [1.55, 4.80] | 3.83 | < .001*** |
| T2$_{(Loss)}$ | -3.24 | [-4.84, -1.64] | -3.97 | < .001*** |

Coefficients (B) represent percentage change in pupil diameter per unit increase of the factor. Subscripted values represent the default value for each factor.

LDW = Loss-Disguised-as-a-Win.

* $p < .050$,

** $p < .010$,

*** $p < .001$.

elicit pupillary contraction that could conceivably cancel out any arousal-related pupil dilation, thus contributing to Type 2 error on those reinforcing outcome types in Experiment 1A. In Experiment 1B, we looked for systematic, outcome-related changes in the average luminance of the screen on genuine slot machines.

## Method

To characterize this possible confounding role of luminance changes, we extracted screen-by-screen luminance values from the slot machine videos captured as part of Experiment 1A. We synchronized these luminance values to the event epochs defined in Experiment 1A. In performing these analyses, we were mindful that any luminance changes could be unique to the colors and associated luminance of the specific slot machine that we used, Buffalo Spirit (see Fig 3). For example, Buffalo Spirit displays a relatively light background for the large 'reel spin' area, and thus the symbols, including those associated with the bonus rounds, tend to be relatively dark. To address this possibility, we extracted screen luminance from archival data from a second authentic slot machine game in our laboratory, "Ice Empress" (Scientific Games Co., Las Vegas, NV). This game is similar to Buffalo Spirit in terms of its structural properties but employs a distinct visual aesthetic in which lighter symbols are presented on a darker screen background (see Fig 3).

The luminance extraction for Ice Empress used a dataset from an unpublished behavioral experiment (Ferrari & Clark. [Unpublished]) in which 31 undergraduate participants played the slot machine for a 20-minute session. This study was conducted with ethical approval from the University's Behavioral Research Ethics Board and written informed consent was taken from all participants. For clarity, we note the present analysis does not use any actual participant data from these sessions; we are solely using video capture of the slot machine's monitor to examine *game-level* characteristics.

**Data analysis.** We modified the software used to produce the time series of slot machine outcomes in Experiment 1A, to convert each frame and pixel of the participants' slot machine screen recordings into the CIELAB color space, separating three channels: L, the lightness (or "luminance") scale, as well as *a* and *b*, the color components. For each frame, we extracted an average luminance (L) value for all pixels, ranging from 0 to 255. CIELAB is preferable to the

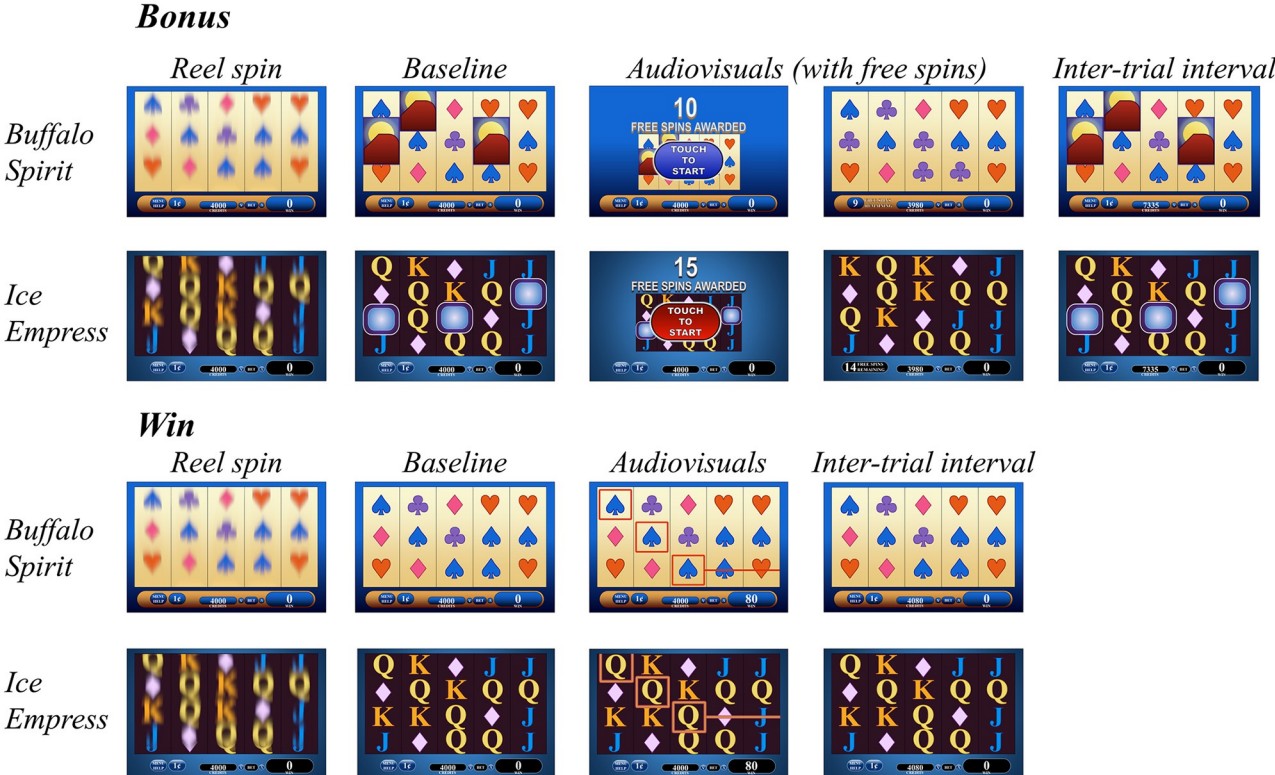

**Fig 3. Illustrations depicting the sequence of on-screen events on free-spin bonus features and winning outcomes, for the two slot machine models used in the present study.** These illustrations capture the approximate colour schemes and image dimensions of the key game elements. Researchers requiring actual screenshots should contact the corresponding author. Losses-disguised-as-wins follow similar in-game processes to the winning outcomes shown, and loss outcomes are similar to wins during the reel spin but do not contain any audiovisual feedback. Note the contrasting luminance properties for Buffalo Spirit (as a light screen background) and Ice Empress (with a darker screen background).

RGB color space because it is perceptually uniform: changes in L correspond proportionally to changes in perceived screen brightness. Analyses were conducted using R (4.0.0).

We calculated these average L values as a function of the four outcome types in Experiment 1A, for all participants and all trials. For exploratory purposes, we plot these luminance levels for the full duration of the three phases of each 'trial' (gamble): the reel spin (i.e., prior to the outcome), the audiovisual feedback, and the subsequent inter-trial interval (Fig 4). Trial duration varied both between but also within outcome types, such that large error bars are evident towards the end of the three phases, due to the diminishing number of trials. We discarded frames exceeding that of the mean plus one standard deviation of trial duration, separately for each outcome by epoch. To facilitate a more direct comparison with Experiment 1A, we also plotted L values within the shorter epochs reflecting the Baseline, T1 and T2 phases (Fig 4); these epochs fall within the longer reel spin, audiovisual feedback, and inter-trial interval epochs, as depicted in Fig 1. The Baseline epochs were extracted from the reel spin phase prior to discarding frames, so that the Baseline would be immediately 'contiguous' with the T1 epoch; for this reason, the Baseline and reel spin plots do not perfectly align in Fig 4.

## Results

**Buffalo spirit.** A number of interesting features are evident in the full time-courses depicted in Fig 4. First, during the reel spin, bonus rounds are associated with gradual decrease

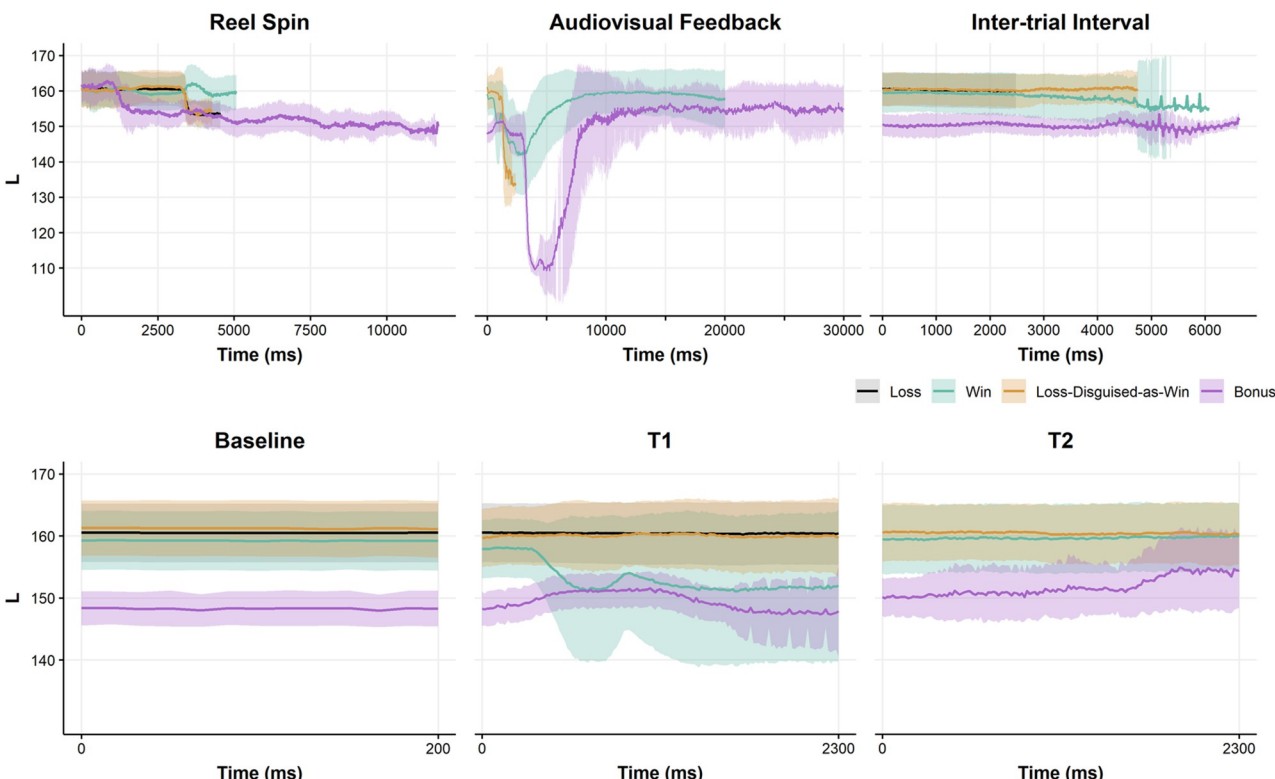

**Fig 4. Buffalo spirit slot machine screen luminance by outcome phase.** Average screen luminance (L) component from CIELAB color space time-locked to event onset. L range = 0–255, Audiovisual Feedback Time ranged from 0–140,430 ms; a narrower range limited at 30,000 ms is presented for easier inspection and comparisons to T1. Ribbons represent one standard deviation above and below the mean of 7284 loss (79.2%), 909 win (9.9%), 942 loss-disguised-as-win (10.2%), and 62 bonus (0.7%) outcomes at a given point in Time. Baseline and Reel Spin outcomes do not align due to discarded frames in the latter.

in screen luminance, which is not evident for losses, wins, and LDWs (see also Table 2). There are also notable differences in the durations of the reel spin (see Fig 4 and S1 Table), such that the spins leading up to the bonus rounds are substantially elongated due to the unique audiovisual accompaniment to each bonus symbol 'landing' [67]. The bonus symbols are roughly

**Table 2. Averaged slot machine screen luminance, full epochs.**

|  | N | Reel Spin | Audiovisual Feedback | Inter-trial Interval |
|---|---|---|---|---|
| *Buffalo Spirit* |  |  |  |  |
| Loss | 7284 | 158.80 (4.65) | — | 160.23 (4.87) |
| Win | 909 | 159.94 (4.98) | 156.15 (8.09) | 158.04 (7.93) |
| LDW | 942 | 159.38 (4.58) | 149.27 (7.18) | 160.43 (4.53) |
| Bonus | 62 | 152.88 (3.92) | 150.10 (9.49) | 150.19 (2.90) |
| *Ice Empress* |  |  |  |  |
| Loss | 3060 | 76.49 (6.95) | — | 74.25 (7.06) |
| Win | 406 | 78.01 (8.30) | 85.96 (7.03) | 81.64 (8.16) |
| LDW | 292 | 76.69 (6.95) | 77.21 (6.52) | 79.05 (6.71) |
| Bonus | 36 | 79.37 (5.73) | 79.69 (9.87) | 80.63 (5.37) |

LDW = Loss-Disguised-as-a-Win.

twice as large as the symbols that award regular wins, they are darker, and as they appear successively on three (or more) of the five reels (Fig 3), this likely accounts for the apparent incremental decrease in luminance.

Second, during the audiovisual feedback phase, a large dip in luminance was apparent on bonus rounds (about 20% of the full range of the luminance scale, or ≈50L out of 255). Bonus features are unique in that once triggered, the reels shrink, and a large dark blue border alerts the user to the free spins (Fig 3). This effect is the presumed source of the luminance dip. As the bonus spins begin, the reels return to full size, and there is a corresponding recovery in luminance to its previous level. Third, a luminance dip is also apparent following wins (about 10% of the full range of the luminance scale, or ≈25L out of 255), with a faster onset than the dip on bonuses. This likely represents the appearance of the flashing, colored borders around the winning paylines (Fig 3), which tend to be darker in color, given the lighter screen background of Buffalo Spirit. Lastly, during the inter-trial interval phase, the reels reset to the original winning combination of symbols prior to audiovisual onset (Fig 3). Thus, L at the inter-trial interval and baseline epochs are equivalent, and as such, the lower L on bonus rounds recurs during the inter-trial interval. As the machine is still during the inter-trial interval, the L values are stable throughout this phase for each outcome type (Fig 4).

The luminance fluctuations described above may impinge on the shorter epochs from Experiment 1A. As the Baseline epoch was defined as the final 200 ms of the reel-spin, the L at Baseline is lower on bonuses compared to the other three outcome types (Table 3), presumably due to the visual properties of the bonus symbols. During T1, the wins diverge from losses and losses-disguised-as-wins, drawing somewhat nearer to bonuses, which remained lower on average (Table 3). During T2, wins and losses-disguised-as-wins appeared to have a slightly higher average luminance than bonuses (Table 3).

**Ice empress.** Ice Empress employs a darker background than Buffalo Spirit (Fig 3), and thus the overall screen luminance is considerably lower (≈75L) compared to Buffalo Spirit (≈150L). On Ice Empress, there is minimal evidence of luminance changes during the reel spin or the inter-trial interval phase (see Fig 5 and Table 2). However, during the audiovisual feedback phase, the bonus rounds were associated with a large spike in L (i.e., a rapid increase and recovery), which was also evident to a smaller extent following wins, and thus *mirrored* the luminance dips observed on Buffalo Spirit, as a game that uses a lighter theme. Ice Empress bonuses follow the analogous structure as Buffalo Spirit, but with the color scheme inverted such that the signaling border change is light relative to the reels (Fig 3), leading to a spike in luminance.

## Discussion

Slot machine outcomes were associated with systematic and multi-faceted changes in screen luminance, most notably as a sustained effect during the audiovisual feedback that

**Table 3. Buffalo spirit averaged slot machine screen luminance, measured epochs.**

|  | *N* | **Baseline** | **T1** | **T2** |
|---|---|---|---|---|
| Loss | 7284 | 160.50 (4.74) | 160.42 (4.94) | — |
| Win | 909 | 159.25 (4.76) | 153.10 (10.20) | 159.67 (5.52) |
| LDW | 942 | 161.23 (4.51) | 160.06 (5.44) | 160.47 (4.74) |
| Bonus | 62 | 148.27 (2.77) | 149.43 (3.47) | 151.75 (4.68) |

LDW = Loss-Disguised-as-a-Win.

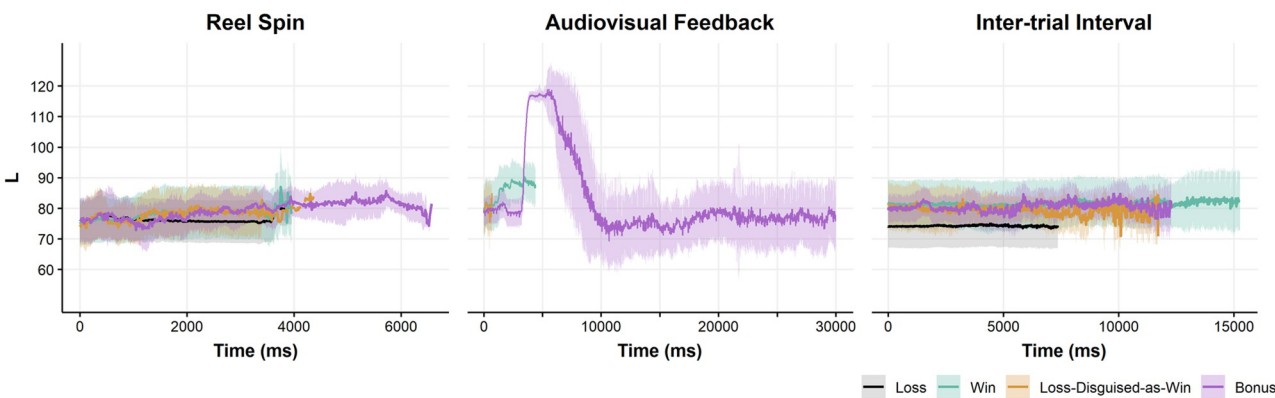

**Fig 5. Ice Empress slot machine screen luminance by outcome phase.** Average screen luminance (L) component from CIELAB color space. L range = 0–255, Audiovisual Feedback Time ranged from 0–176,220 ms; a narrower range limited at 30,000 ms is presented for easier inspection. Ribbons represent one standard deviation above and below the mean of 3060 loss (82.0%), 406 win (10.9%), 229 loss-disguised-as-win (6.1%), and 36 bonus outcomes (1.0%) at a given point in Time.

accompanies bonus features and, to a lesser extent, winning outcomes. These fluctuations were relatively modest during the shorter windows (i.e., Baseline, T1, T2 epochs) from which pupil diameter was extracted in Experiment 1A, and within those shorter epochs, luminance was quite stable (Fig 4). Thus, although large-scale luminance fluctuations occurred during these two slot machine games, in our view there is not a clear 'luminance-based' explanation for the significant effects on pupil diameter that we observed in Experiment 1A in response to the bonus features (see General Discussion for further discussion).

One limitation pertaining to these analyses of screen luminance derived from video capture is that objective 'on screen' luminance is not the same as the brightness perceived by our study participants, which is affected by luminous flux incident on the eye, and subject to further variability as a function of eye movements [7]. The key question in Experiment 2 is whether the pronounced luminance changes characterized in the screen extraction analysis are sufficient to cause pupillary responses. This will clarify the role of luminance as a confound to reward-related pupillometry measures.

## Experiment 2

### Method

**Stimulus preparation.** We created a stimulus set using scrambled screen shots from our two slot machines, taken at the moment of peak luminance fluctuations that were observed on wins (≈25L decrease on Buffalo Spirit, ≈25L increase on Ice Empress) and bonus rounds (≈50L decrease on Buffalo Spirit and ≈50L increase on Ice Empress). We created two further stimuli intended as manipulation checks, with a greater ≈100L luminance difference (decrease on Buffalo, increase on Ice Empress) to ensure the sensitivity of our pupillometry measurement. We presented these image stimuli to new participants during pupillary recording, in a laboratory study that was a hybrid of a visual psychophysics design but displayed on a slot machine monitor to equate the screen properties, seating distance, and ambient lighting with Experiment 1A.

Specifically, our stimulus set comprised six conditions: three luminance contrasts for each of the two slot machines (see Fig 6). The stimuli were constructed from screen shots of the two slot machine games, which were scrambled to ensure that the displayed events would

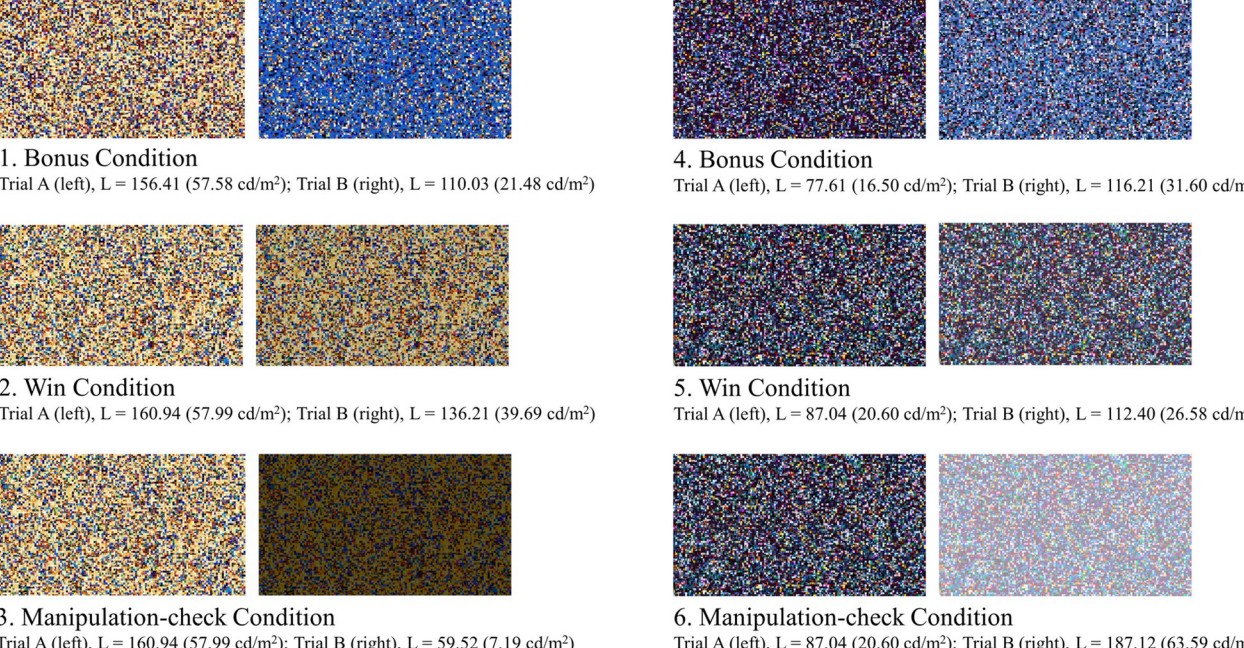

**Fig 6. Scrambled images of slot machine outcomes.** L = objective luminance, cd/m2 = photometer measured brightness rating in situ ambient lighting.

not be perceived as exciting or financially rewarding. Luminance within these screenshots was manipulated using Adobe Photoshop (v.21.1.10). Each condition comprised a 5s baseline stimulus, henceforth termed Trial A, based on the screen prior to the key game event, and then a 5s stimulus, henceforth Trial B, that captured the peak luminance changes from the screen luminance analysis. The conditions were as follows: 1) Bonus Conditions, for which Trial B depicted a 50L decrease in the Buffalo Spirit, and a 50L increase in the Ice Empress condition; 2) Win conditions, for which Trial B depicted a 25L decrease in the Buffalo Spirit, and a 25L increase in the Ice Empress condition. Because the visual stimulation that accompanies wins varies in color, size, and patterns, the Trial B stimulus was created directly from stimulus A, using Photoshop. 3) The manipulation-check conditions, in which Trial A was identical to the win condition, and Trial B was the same as Trial A, but manipulated by ≈100L.

**Participants and procedure.** We recruited 15 graduate students and university staff from the UBC Psychology Department (9 men and 6 women; mean age = 23.4, *SD* = 3.3), using the same exclusion criteria as in Experiment 1A. Additional inclusion criteria imposed by running the study during the COVID-19 pandemic was that participants could not be over 60 years of age, immunocompromised, and/or suffering from chronic diseases. This experiment used a different eye tracker than Experiment 1A, a decision that was enforced by SMI discontinuing product support. We used a mobile eye-tracker (Pupil Labs, Berlin, Germany), that is very similar to the SMI glasses with the exception that prescription lenses are not available. As such, participants with corrected-to-normal vision needed to wear contact lenses to avoid recording artifacts due to different pairs of glasses. Three participants were excluded due to equipment

error. In one other participant, pupillary recording was lost after 16 of 30 blocks but the available data was valid and included. Thus, analyses are reported on 12 participants.

After giving informed consent, participants were seated in front of an authentic slot machine chassis wearing the eye tracking equipment. The sessions began with a 5-minute dark adaptation. The task was presented via PowerPoint (v.17.0) and fed through an HDMI cable to appear on the slot machine screen, a 22" LCD monitor (Wells-Gardner WGF2298A, McCook, IL). We employed a colorimeter (ColorCAL MKII, Cambridge Research Systems, Kent, UK) positioned at a viewing distance of 60cm from the slot machine screen to confirm the correspondence between our stimuli's screen luminance (L) and the luminance of those stimuli emitted from the monitor ($cd/m^2$) (see Fig 6). Additionally, the monitor emitted a maximum luminance rating of 129.25 $cd/m^2$, and the *in situ* ambient lighting conditions of the laboratory was minimal (0.13 $cd/m^2$). Other than the slot machine, the lighting conditions of the testing room were constant.

The six conditions were presented in blocks that comprised four repetitions of Trial A and Trial B, followed by a 30 s pupil recovery time in the dark after the offset of the fourth repetition (for comparable protocols, see [68, 69]). Each condition was presented 5 times, creating 30 blocks. To minimize eye movements, a small, white-dotted fixation point appeared in the center of the screen 0.5 s prior to the offset of a stimulus and disappeared with the onset of the next stimulus. Condition order was balanced using a Latin square design.

**Data analysis.** Pupil Lab's Pupil Capture software extracted artefacts such as pupil detection ("confidence") and pupil diameter from videos at 60 frames per second. High confidence values (max 1.0) indicated high quality of pupil detection for a given frame. Data points with confidence of less than .80 indicated potential blinks and saccades, and were discarded (5.93% of datapoints from Buffalo Spirit; 6.59% from Ice Empress). Epochs contained enough observations to calculate means, and thus missing data were not interpolated (i.e., missing data were treated as NaNs in the analysis).

Analyses were conducted separately for Buffalo Spirit and for Ice Empress using R (4.0.0). We calculated the mean pupil diameter in the last 200 ms of Trial A, and the mean pupil diameter in an interval from 200 ms to 2,500 ms on Trial B (see Fig 7), and our measure of pupil diameter change was the difference score from Trial B minus Trial A. Our defined time periods were in keeping with Experiment 1A and favorable to alternatives. For example, were we to calculate difference scores using two contiguous 2.5 s epochs, the latter 2.5 s of Trial A allows for gradual increases in pupil diameter. Specifically, pupils 'escape' from sustained light or 'recover' from darkness [5], and such events might occur, respectively, for Buffalo Spirit (where Trial A is lighter than Trial B) and Ice Empress (Trial A is darker than Trial B). Hence, the last 200 ms at Trial A occurring immediately prior to Trial B provides an appropriate time period of comparison. We aggregated the difference scores across the 5 (or fewer, for one participant) repetitions of each condition. We tested for trends across the sequence within each block, and these effects were non-significant and so we proceeded with the analysis on the aggregated difference scores.

Tests of univariate normality (Shapiro-Wilk) revealed residual non-normality ($p < .05$). Thus, as an initial test of whether difference score medians were significantly different from zero, we conducted the Wilcoxon signed-rank test, a non-parametric alternative to the sample *t*-test. Effect sizes were estimated using $R^2$ as an alternative to Cohen's *d* [70]. Mauchly's test yielded non-significant results ($p > .05$), and thus no correction for non-sphericity was applied. Because ANOVA is generally robust to violations of normality [71], we did not correct for non-normality in our ANOVAs.

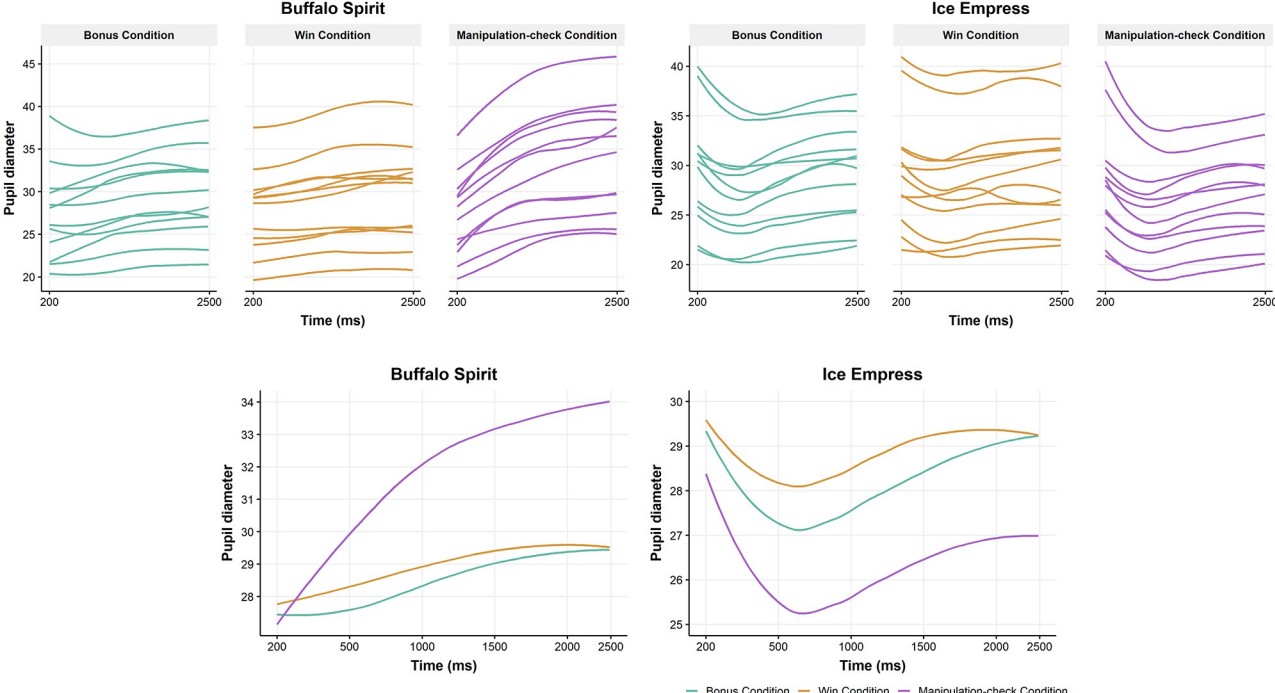

**Fig 7. Pupil time-course.** In the top row, lines indicate the mean pupil diameter from 200 to 2500 ms on Trial B for each participant (n = 12). In the bottom row, lines indicate the mean pupil diameter from 200 to 2500 ms on Trial B aggregated across participants. Pupil diameter is estimated as pixels as observed in the eye on the camera image, and is therefore not corrected for perspective.

## Results

On Buffalo Spirit, there was a significant effect of luminance condition on the pupil diameter difference scores, $F(2, 22) = 65.88$, $p < .001$. Pairwise tests (Bonferroni correction to $\alpha = .017$) found that the $\approx$100L manipulation check stimulus pair was associated with significantly greater pupil changes than the bonus ($\approx$50L) and win ($\approx$25L) stimulus pairs, which did not differ from one another. Testing each stimulus pair against a reference value of 0% change, the pupillary response was significant for each stimulus pair, with medium to large effect sizes ($R^2$ = .28 to .76; Table 4).

**Table 4. Difference score medians, IQRs, Wilcoxon signed-rank test p-values and effect sizes.**

|  | Median | IQR | $p$ | $r$ | $R^2$ |
|---|---|---|---|---|---|
| *Buffalo Spirit* |  |  |  |  |  |
| Bonus Condition | .78 | .90 | < .001 | .55 | .30 |
| Win Condition | .66 | 1.34 | < .001 | .53 | .28 |
| Manipulation-check Condition | 4.26 | 2.02 | < .001 | .87 | .76 |
| *Ice Empress* |  |  |  |  |  |
| Bonus Condition | -3.11 | 2.28 | < .001 | .87 | .76 |
| Win Condition | -1.87 | 1.97 | < .001 | .87 | .76 |
| Manipulation-check Condition | -4.40 | 1.94 | < .001 | .87 | .76 |

$N = 12$.

The equivalent model for Ice Empress also found a significant effect of luminance conditions, $F(2, 22) = 62.42$, $p < .001$. Follow-up pairwise tests revealed that the manipulation condition produced significantly greater pupil changes than the bonus condition, and the win condition, and there was also a significantly greater response to the bonus condition compared to the win condition. The difference score medians per condition, representing pupil contraction (i.e., decreases in Trial B relative to Trial A), were each significantly less than the population median of zero, with large effects ($R^2 = .76$; Table 4).

## Discussion

Although the pupillary changes were clearly of greatest magnitude in the manipulation-check condition, the two stimulus pairs that were based on realistic in-game luminance fluctuations were sufficient to trigger changes in pupil diameter. This highlights a need to account for incidental luminance changes in future studies employing ecologically-valid gambling products.

## General discussion

We investigated physiological correlates of genuine slot machine gambling using pupillary measures obtained from mobile eye tracking glasses. In Experiment 1A, we found significant pupil dilation (a marker of sympathetic arousal) immediately after participants received free spin bonus feature outcomes. In Experiment 1B, we noted differences in luminance between slot machine outcomes as a potential confound to our arousal-related effects. In Experiment 2, luminance fluctuations using scrambled slot machine displays and scaled on the actual game-related changes were sufficient to trigger reliable pupil responses.

### Do pupillary responses to bonus rounds reflect sympathetic arousal or a reflexive response to lighting changes?

We observed significant pupil dilation in response to receiving free spin bonus features in Experiment 1A. There was no evidence that pupil diameter was sensitive to reinforcement from wins and losses-disguised-as-wins (relative to losses). The rarer, more-salient bonus features may produce a greater sense of excitement than conventional winning outcomes (and losses-disguised-as-wins), to which gamblers may rapidly habituate. By this account, modern slot machines might not be *predominantly* physiologically arousing; rather, their appeal could be based on the pursuit of occasional events such as bonus rounds or larger jackpot wins, or on providing interesting or immersive gameplay features.

The significant increase in pupil diameter for bonus features at T1 relative to baseline did not evidently co-occur with any decrease in screen luminance for bonuses at T1. Moreover, luminance values for the different slot machine outcomes varied little from baseline to T1 to T2. Thus, although we found marked changes in luminance during the reinforcing outcomes (discussed in the next section), such changes occurred outside the time course of our measured epochs in Experiment 1A. This favors an arousal- or excitement-based account of the pupillary effect to bonus features, rather than a reflexive response to incidental changes in screen brightness.

Nevertheless, pupil diameter can be influenced by several factors. For example, the duration of the reel spin was substantially longer during free spin bonus features than other outcomes (see Fig 4 and S1 Table). Because pupil diameter is sensitive to uncertainty and anticipation of reward [72, 73], it is unclear from our findings whether the effects of bonus features are due to reward *per se*, or due to the anticipation of reward. We also note that the symbols that trigger bonus rounds on 'Buffalo Spirit' are physically larger and accompanied by loud and distinctive auditory flourishes. Recent research has shown that pupil dilation can be elicited by auditory

and/or visual stimuli, and further modulated by stimulus salience [74]. Thus, the observed effects of bonus rounds on pupil size may be due to their auditory and visual features, as well as their relative salience and infrequency. In contrast to bonuses, loss outcomes occurred on around 80% of trials, and this relative predictability is less than ideal for using the losses as the reference category for all three types of reinforcing outcomes. The low frequency of bonus rounds also means that these events occurred in only a subset of sessions and participants. Because we used a genuine slot machine, we could not control these outcome frequencies, and our analyses involving bonus features draw upon a very small number of observations (see S1 Table). This likely diminished the reliability of the reported coefficient estimates.

## What game events trigger luminance fluctuations in authentic slot machines?

We observed luminance fluctuations during the delivery of wins and bonuses, in Buffalo Spirit (decreasing luminance; Fig 4) and Ice Empress (increasing luminance; Fig 5). In line with the pupil light reflex [5], the onset of darker stimuli on a slot machine monitor increased pupil diameter (Buffalo Spirit; Table 4) and the onset of brighter stimuli decreased pupil diameter (Ice Empress; Table 4). These data suggest that meaningful on-screen luminance changes accompany wins and bonuses, and therefore that caution is required in attributing the pupillary effects on positively-reinforcing outcomes to the sympathetic nervous system. Nonetheless, as the luminance fluctuations on bonuses occurred after T1, it is unlikely that the pupil dilation observed at the onset of the bonus audiovisuals (T1) was related to changes in screen luminance.

The strength of the pupil response scales with the magnitude of luminance change: the manipulation-check condition that offered the greatest luminance difference ($\approx$100L) produced significantly greater pupil response than the two stimulus pairs derived from the slot machine game, at $\approx$25L for wins and $\approx$50L for bonuses. Further, the pupil response appears to be more sensitive to slot machine screen brightness than darkness. Specifically, the bonus condition produced stronger effects than the win condition in Ice Empress (but not in Buffalo Spirit). This is consistent with research suggesting that pupils constrict more immediately to brightness and recover more gradually in darkness [5]. It is likely that slot machine manufacturers use luminance to increase the saliency of positively-reinforcing outcomes. The practical implication of our work is that future slot machine research utilizing pupillometry should consider luminance fluctuations in the context of the specific gambling product being used, and account for luminance in terms of both magnitude and direction of change.

There is increasing interest in the use of pupillometry and other eye tracking metrics in relation to gambling [75]. Refining techniques that can separate the pupil effects of cognitive activity vs. light input in realistic settings (e.g., Marshall's Index of Cognitive Activity; [76–78]) would further enhance the use of pupillometry for indexing surprise, excitement or other emotional and motivational processes that are elicited during slot machine gambling [79]. This would add to the growing body of work that examines physiological states and how individual differences in psychophysiological reactivity relate to risk of gambling harm.

## Limitations

Experiment 1A sought to examine slot machine gambling in an ecologically-valid manner. We could not control any perceptual or phenomenal differences in slot machine events or areas of the game screen, and we did not correct for individual differences in pupillary foreshortening [80]. The hardware differences between the two studies may have differentially impacted foreshortening error (see S1 Text). On bonus features, unique audiovisual features occurred

during the baseline pupil measurement, such that this epoch is unlikely to represent a neutral baseline. In contrast, Experiment 2 was more tightly controlled, and the stimulus pairs were chosen to capture the peak luminance effects, but those observations may not generalize to real-life slot machine gambling, which contains other confounding factors besides luminance (e.g., reinforcement schedules, slot machine sounds, ambiance of the casino). Several additional limitations are discussed in the S1 Text.

## Conclusion

Mobile eye tracking technology offers a valuable tool for exploring gamblers' reactions to gambling products and environments. Our results suggest that reinforcement from slot machine bonus rounds is associated with pupillary dilation, but that future studies should pay careful attention to fluctuations in luminance as a possible confound in the interpretation of pupillary effects. Researchers should expect that these fluctuations may be game-specific, and their assessment will require both high resolution video capture and event-related analyses.

## Supporting information

**S1 Text.**
(DOCX)

**S1 Table. Outcome frequencies and durations.** Loss outcomes do not involve any audiovisual feedback and thus we could not specify a T2 event (that was distinct from T1). Bonuses include all free spins during the Audiovisuals phase. LDW = Loss-Disguised-as-a-Win.
(DOCX)

**S1 Fig. Correlation between corrected and uncorrected pupil diameter.** Within-subjects correlations for each of the 53 participants in Experiment 1 are shown. Data labels depict the maximum, median, and minimum correlations.
(TIF)

## Author Contributions

**Conceptualization:** Andy J. Kim, W. Spencer Murch, Luke Clark.

**Data curation:** Andy J. Kim, W. Spencer Murch.

**Formal analysis:** Andy J. Kim, W. Spencer Murch.

**Funding acquisition:** Luke Clark.

**Investigation:** Andy J. Kim, W. Spencer Murch.

**Methodology:** Andy J. Kim, W. Spencer Murch, Luke Clark.

**Project administration:** Andy J. Kim, W. Spencer Murch.

**Resources:** W. Spencer Murch, Luke Clark.

**Software:** Andy J. Kim, W. Spencer Murch, Eve H. Limbrick-Oldfield, Mario A. Ferrari, Kent I. MacDonald.

**Supervision:** W. Spencer Murch, Luke Clark.

**Validation:** Andy J. Kim, W. Spencer Murch, Eve H. Limbrick-Oldfield.

**Visualization:** Andy J. Kim, W. Spencer Murch.

**Writing – original draft:** Andy J. Kim, W. Spencer Murch.

**Writing – review & editing:** Andy J. Kim, W. Spencer Murch, Eve H. Limbrick-Oldfield, Mario A. Ferrari, Kent I. MacDonald, Jolande Fooken, Mariya V. Cherkasova, Miriam Spering, Luke Clark.

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
