## [Decision Letter · Decision Letter 0]

28 Feb 2022

PONE-D-22-00740Do pupillary responses during authentic slot machine use reflect arousal or screen luminance fluctuations? A proof-of-concept studyPLOS ONE

Dear Dr. Kim,

Thank you for submitting your manuscript to PLOS ONE. After careful consideration, we feel that it has merit but does not fully meet PLOS ONE’s publication criteria as it currently stands. Therefore, we invite you to submit a revised version of the manuscript that addresses the points raised during the review process.

Please address all comments made by the reviewers comprehensively.

We look forward to receiving your revised manuscript.

Kind regards,

Manuel Spitschan

Academic Editor

PLOS ONE

Journal Requirements:

(I have read the journal's policy and the authors of this manuscript have the following competing interests: 

The Centre for Gambling Research at UBC receives funding from the Province of British Columbia and the British Columbia Lottery Corporation (BCLC), a Canadian Crown Corporation. The slot machines used in the present study were provided to the Centre for Gambling Research by the BCLC. 

EHLO has received a speaker honorarium from the Massachusetts Council on Compulsive Gambling (USA). She has accepted travel or accommodation for speaking engagements from the National Council for Responsible Gambling (USA), the International Multidisciplinary Symposium on Gambling Addiction (Switzerland), and the Responsible Gambling Council (Canada). She has not received any further direct or indirect payments from the gambling industry or groups substantially funded by gambling.

MAF has received a speaker honorarium from the British Columbia Lottery Corporation (BCLC). 

MVC has received a speaker honorarium from the Responsible Gaming Association of New Mexico (USA).

LC is the Director of the Centre for Gambling Research at UBC. LC has received speaker/travel honoraria from the National Association of Gambling Studies (Australia) and the International Center for Responsible Gaming (USA). He has received academic consulting fees from Gambling Research Exchange Ontario (Canada), GambleAware (UK), and the International Center for Responsible Gaming (USA). He has not received any further direct or indirect payments from the gambling industry or groups substantially funded by gambling. He has received royalties from Cambridge Cognition Ltd. relating to neurocognitive testing.

AJK, WSM, KIM, JF and MS report no conflicts of interest.)

3. We note that you have referenced) (Ferrari & Clark, unpublished) which has currently not yet been accepted for publication. Please remove this from your References and amend this to state in the body of your manuscript: (ie “Bewick et al. [Unpublished]”) as detailed online in our guide for authors

4. We note that Figure 3 in your submission contain copyrighted images. All PLOS content is published under the Creative Commons Attribution License (CC BY 4.0), which means that the manuscript, images, and Supporting Information files will be freely available online, and any third party is permitted to access, download, copy, distribute, and use these materials in any way, even commercially, with proper attribution. For more information, see our copyright guidelines: http://journals.plos.org/plosone/s/licenses-and-copyright.

a. You may seek permission from the original copyright holder of Figure 3 to publish the content specifically under the CC BY 4.0 license. 

Reviewers' comments:

Reviewer's Responses to Questions

**Comments to the Author**

1. Is the manuscript technically sound, and do the data support the conclusions?

Reviewer #1: Yes

Reviewer #2: No

2. Has the statistical analysis been performed appropriately and rigorously? 

Reviewer #1: No

Reviewer #2: Yes

3. Have the authors made all data underlying the findings in their manuscript fully available?

Reviewer #1: Yes

Reviewer #2: Yes

4. Is the manuscript presented in an intelligible fashion and written in standard English?

Reviewer #1: Yes

Reviewer #2: Yes

5. Review Comments to the Author

Reviewer #1: Do pupillary responses during authentic slot machine use reflect arousal or screen luminance fluctuations? A proof-of-concept study

The manuscript explores whether pupillometry can give insight into arousal in a slot machine gambling scenario and asks whether arousal-linked pupil responses can be differentiated from those caused by intense audiovisual feedback. The task is ecologically valid--real slot machines were used--and participants (n = 53) were themselves slot machine gamblers. In the first experiment, pupil data related to wins, losses-disguised-as-wins, and bonus feature events were compared to a loss baseline. The pupil response was greatest for bonus feature events, which were characterised by intense audiovisual feedback. In the second part of the first experiment, real-time luminance data are extracted from the displays. Analyses and visualisations of these data confirmed that there are systematic effects of luminance between the different events analysed in the first part of the experiment. Finally, experiment 2 used scrambled stills of the slot machine events to confirm that pupil responses were indeed triggered by luminance information.

The writing is clear, easy to follow, and close to publication standard. I have the following comments / suggestions:

Comments / suggestions

1. It is interesting to see cognitive pupillometry applied in the context of a slot machine gambling task. I feel that the narrative could be enhanced if the authors were able to place their findings within the wider context of studies that attempt to use pupillometry in realistic settings. For example, what do the author's think of Marshall's (2008) index of cognitive activity (which purports to factor out luminance changes), and Bhavsar et al.'s control room operator study?

Marshall, S. P. (2002). The Index of Cognitive Activity: Measuring cognitive workload. Proceedings of the IEEE 7th Conference on Human Factors and Power Plants, 7-5-7–9. https://doi.org/10.1109/HFPP.2002.1042860

Bhavsar, P., Srinivasan, B., & Srinivasan, R. (2016). Pupillometry based real-time monitoring of operator’s cognitive workload to prevent human error during abnormal situations. Industrial and Engineering Chemistry Research, 55(12), 3372–3382. https://doi.org/10.1021/acs.iecr.5b03685

2. In Experiment 2 (p20), the author's state 'Data points with confidence of less than .80 indicated potential blinks and saccades, and were discarded (5.93% of datapoints from Buffalo Spirit; 6.59% from Ice Empress). Epochs contained enough observations to calculate means, and thus missing data were not interpolated.' -- were the discarded data points treated as NaNs? It would be interesting to see the time-courses in addition to the summary statistics presented in Table 4.

3. The main conclusion of the discussion (p25) states, 'The practical implication of our work is that future slot machine research utilizing pupillometry should account for luminance in terms of magnitude and direction of change.'. I feel that this statement is rather obvious, and that it applies more generally to any research where cognitive pupillometry is used in real life scenarios (see first comment). I therefore think the authors should comment more on why they think this is important -- what exactly is the promise, here? If there was a perfect system that could disentangle luminance- and arousal-related pupil responses, what more could we learn about slot machine gambling behaviour?

4. The author's admit that they did not correct for pupil foreshortening. Was the risk of this different for the two eye trackers that were used? I know that the Pupil Core system uses a 3-D model of pupil size that is robust to the effects of gaze position at the population level (Petersch & Dierkes, 2021), but I'm not sure about the SMI system. Some more info here would be useful. Also, as no corrections were applied, and as the overall findings are not particularly striking, I think the manuscript would benefit from further analyses to explore whether pupil size was systematically related to gaze position.

Petersch, B., & Dierkes, K. (2021). Gaze-angle dependency of pupil-size measurements in head-mounted eye tracking. Behavior Research Methods. https://doi.org/10.3758/s13428-021-01657-8

Brisson, J., Mainville, M., Mailloux, D., Beaulieu, C., Serres, J., & Sirois, S. (2013). Pupil diameter measurement errors as a function of gaze direction in corneal reflection eyetrackers. Behavior Research Methods. https://doi.org/10.3758/s13428-013-0327-0

Martin, J. T., Whittaker, A. H., & Johnston, S. J. (2020). Component processes in free-viewing visual search: Insights from fixation-aligned pupillary response averaging. Journal of Vision, 20(7), 5. https://doi.org/10.1167/jov.20.7.5

Reviewer #2: The study examines pupillary responses during authentic slot machine. The results show some differences in pupil size in some comparisons, and these observed differences in pupil size cannot be simply explained by the differences of overall luminance level. Overall, I think this study is interesting and the motivation is clear, particularly it is important to study in the real-world gambling situation. However, there are a range of factors that also influence pupil size, and differences in visual, auditory and temporal features between different conditions may have confounded the observed results.

My major concern comes with the feedback differences in visual and audio features between 4 conditions. If I understand correctly, the critical time epoch should be the differences in pupil size between T1 and T2 (feedback epoch), as any results after T2 could overlap with the next run temporally. However, feedback durations are different among different conditions, with longest for wins, shortest for LDWs, no feedback for losses. First, it is unclear the duration “range” of the feedback in these 4 conditions, also it is not clear about bonus feedback. So, it is better to clearly describe this information in Table (duration range). Furthermore, the main comparison is between the win/LDW/bonus and loss conditions. If there is no audiovisual feedback in the loss condition, the T1-T2 duration for the loss condition is likely overlapping with the next run temporally, how to perform this comparison properly? Thus, if the win and loss (or LDW) conditions have a relatively similar trial features (e.g., duration), it may be better to compare these two conditions only.

The authors conducted some other experiments to control luminance influence on pupil size, which is appropriate. However, there are many other visual factors that also influence pupil size (e.g., Barbur, J. L. (2004). Learning from the pupil-studies of basic mechanisms and clinical applications. The visual neurosciences, 1, 641-656). Besides, pupil dilation is observed after acoustic sounds, and the magnitude of dilation is scaled with stimulus intensity (also arousing level). All these factors need to be taken into consideration to make sure that the observed differences in pupil size are not simply mediated by these differences.

Also, trial number also radically different among different conditions, as most trials probably are loss trials (~60-70 %). Does trial number could affect the observed effects in pupil size?

6. PLOS authors have the option to publish the peer review history of their article (what does this mean?). If published, this will include your full peer review and any attached files.

Reviewer #1: **Yes: **Joel T. Martin

Reviewer #2: No

---

## [Author Response · Author response to Decision Letter 0]

19 May 2022

Date: 2022-May-06

Ms. No.: PONE-D-22-00740

Academic Editor: Dr. Manuel Spitschan

Journal: PLOS ONE

Title: Do pupillary responses during authentic slot machine use reflect arousal or screen luminance fluctuations? A proof-of-concept study

We would like to thank the Academic Editor and Reviewers for their helpful feedback. We have worked to revise our submission in light of the comments and suggestions provided, and we believe the revised manuscript has benefitted substantially from the feedback.

Response to Reviewer 1’s Comments

1. It is interesting to see cognitive pupillometry applied in the context of a slot machine gambling task. I feel that the narrative could be enhanced if the authors were able to place their findings within the wider context of studies that attempt to use pupillometry in realistic settings. For example, what do the author's think of Marshall's (2008) index of cognitive activity (which purports to factor out luminance changes), and Bhavsar et al.'s control room operator study?

RESPONSE: These are good suggestions. We have added the following text in the Discussion to enhance our narrative (the text below also addresses the reviewer’s comment #3). 

“There is increasing interest in the use of pupillometry and other eye tracking metrics in relation to gambling (Li et al., 2021). Refining techniques that can separate the pupil effects of cognitive activity vs. light input in realistic settings (e.g., Marshall’s Index of Cognitive Activity; Bhavsar et al., 2016; Marshall, 2002; Vogels et al., 2018) would further enhance the use of pupillometry for indexing surprise, excitement or other emotional and motivational processes that are elicited during slot machine gambling (Preuschoff, Hart, & Einhäuser, 2011). This would add to the growing body of work that examines physiological states and how individual differences in psychophysiological reactivity relate to risk of gambling harm.” (Page 25-26)

 2. In Experiment 2 (p20), the author's state 'Data points with confidence of less than .80 indicated potential blinks and saccades, and were discarded (5.93% of datapoints from Buffalo Spirit; 6.59% from Ice Empress). Epochs contained enough observations to calculate means, and thus missing data were not interpolated.' -- were the discarded data points treated as NaNs? It would be interesting to see the time-courses in addition to the summary statistics presented in Table 4.

RESPONSE: We apologize for any confusion. To clarify, the discarded data points were treated as NaNs. We have revised the text to state:

“Epochs contained enough observations to calculate means, and thus missing data were not interpolated (i.e., missing data were treated as NaNs in the analysis).” (Page 20)

As for including pupil time-courses, we thank the reviewer for this excellent suggestion. We now include two additional figures (Fig 7) representing pupil time-courses in our manuscript (see below). 

Fig 7. Pupil time-course. In the top row, lines indicate the mean pupil diameter from 200 to 2500 ms on Trial B for each participant (n = 12). In the bottom row, lines indicate the mean pupil diameter from 200 to 2500 ms on Trial B aggregated across participants. Pupil diameter is estimated as pixels as observed in the eye on the camera image, and is therefore not corrected for perspective. 

3. The main conclusion of the discussion (p25) states, 'The practical implication of our work is that future slot machine research utilizing pupillometry should account for luminance in terms of magnitude and direction of change.'. I feel that this statement is rather obvious, and that it applies more generally to any research where cognitive pupillometry is used in real life scenarios (see first comment). I therefore think the authors should comment more on why they think this is important -- what exactly is the promise, here? If there was a perfect system that could disentangle luminance- and arousal-related pupil responses, what more could we learn about slot machine gambling behaviour?

RESPONSE: We agree with the referee, but in our view the impact of luminance fluctuations, and the importance of monitoring and adjusting for these fluctuations, is unlikely to be obvious to the field of gambling studies (e.g. https://www.sciencedirect.com/science/article/pii/S0167876019305434 and preprint http://europepmc.org/article/PPR/PPR478750 ). There is increasing interest in the use of pupillometry and other eye tracking metrics in relation to gambling, but modern gambling products sit in a grey zone between traditional cognitive tasks and modern research conducted in naturalistic settings: slot machines (and other gambling products) represent programmed environments that contain a specified and discrete set of events (wins, losses etc) but these are commercial products where academic researchers typically have minimal access to the underlying code, and so are forced to use ‘naturalistic’ designs. As recommended, we added a section on the impact of our work in the Discussion (Page 25-26): 

“There is increasing interest in the use of pupillometry and other eye tracking metrics in relation to gambling (Li et al., 2021). Refining techniques that can separate the pupil effects of cognitive activity vs. light input in realistic settings (e.g., Marshall’s Index of Cognitive Activity; Bhavsar et al., 2016; Marshall, 2002; Vogels et al., 2018) would further enhance the use of pupillometry for indexing surprise, excitement or other emotional and motivational processes that are elicited during slot machine gambling (Preuschoff, Hart, & Einhäuser, 2011). This would add to the growing body of work that examines physiological states and how individual differences in psychophysiological reactivity relate to risk of gambling harm.”

4. The author's admit that they did not correct for pupil foreshortening. Was the risk of this different for the two eye trackers that were used? I know that the Pupil Core system uses a 3-D model of pupil size that is robust to the effects of gaze position at the population level (Petersch & Dierkes, 2021), but I'm not sure about the SMI system. Some more info here would be useful. Also, as no corrections were applied, and as the overall findings are not particularly striking, I think the manuscript would benefit from further analyses to explore whether pupil size was systematically related to gaze position.

RESPONSE: Thank you. These are valid concerns, and we have thought carefully about how to address this issue in light of the points raised. We attempted the recommended foreshortening correction procedure and generated the results reported in S1 Figure below. However, these efforts revealed a critical problem in the underlying data. Namely, we do not have access to the original raw data for point-of-regard analyses. During the analysis of Study 1, SMI’s take-over by Apple caused them to discontinue academic support, and the output files that we held at that time only included point-of-regard data that had been manually mapped using SMI’s proprietary software package, BeGaze. (This was also the impetus for our switch to a PupilLabs eye tracker for Study 2). As a result, the small proportion of fixations (<1%) that were directed outside of the slot machine screen were mapped to the nearest point on the outer edge of the reference image rather than the location where they truly occurred. As such, our foreshortening correction cannot properly account for fixations occurring away from the slot machine screen, where the largest foreshortening errors would occur.

We state our procedures and explain why we ultimately elected not to correct for pupil foreshortening in S1 Text: 

“The Pupil Core system employed in Experiment 2 uses a three dimensional model of pupil size that is more robust against foreshortening error (Petersch & Dierkes, 2021). Experiment 1 used SMI’s BeGaze analysis software, which provided less robust estimates of pupil diameter. Although participants in that study spent the overwhelming majority of the task staring straight ahead at the slot machine screen (Murch et al., 2020), results in Experiment 1 could have been impacted by participants’ gaze positions during a particular event relative to the position of each eye tracking camera (i.e., foreshortening errors). 

Using available point-of-regard data from Experiment 1, we examined the correlation between uncorrected pupil diameter measurements (in millimeters), and measurements which had been corrected for pupillary foreshortening using a validated linear regression procedure (Brisson et al., 2013; Martin et al., 2020). The individual correlations between corrected and uncorrected pupil diameters are presented in S1 Fig. In general, there was a high correlation between corrected and uncorrected pupil diameters. In the median participant, corrected pupil diameter explained a majority (r2 = 0.73) of the variation in uncorrected pupil diameter, indicating that approximately 27% of variance in pupil diameter was explained by the foreshortening correction model. For comparison, in Brisson et al (2013), 20% of the variance in uncorrected pupil diameter measurements could be explained by the foreshortening-correction regression model.

Crucially, the abrupt closure of SMI in 2017 prevented us from accessing the raw point-of-regard data (prior to our mapping of the data using SMI’s proprietary Semantic Gaze Mapping software), and this impaired the potential utility of our ‘corrected’ pupil diameter measurements. For these data, Semantic Gaze Mapping entailed manually identifying each point-of-regard using a reference image. The borders on this reference image are limited to the size of the slot machine screen, meaning that data pertaining to the visual periphery were lost. As such, our correction models lacked the necessary point-of-regard data needed to fix the most serious instances of pupillary foreshortening. These regression-based foreshortening corrections thus created potentially-misleading measurements for our data. Our analyses are therefore reported without correcting for pupillary foreshortening.”

S1 Fig. Correlation between corrected and uncorrected pupil diameter. Within-subjects correlations for each of the 53 participants in Experiment 1 are shown. Data labels depict the maximum, median, and minimum correlations. 

The reviewer also raises a good point that the risk of foreshortening bias differs between the two experiments (based on the different eye tracking hardware), and we refer to S1 Text in our Limitations (page 26):

“We could not control any perceptual or phenomenal differences in slot machine events or areas of the game screen, and we did not correct for individual differences in pupillary foreshortening [80]. The hardware differences between the two studies may have differentially impacted foreshortening error (see S1 Text).”

Response to Reviewer 2’s Comments

1. My major concern comes with the feedback differences in visual and audio features between 4 conditions. If I understand correctly, the critical time epoch should be the differences in pupil size between T1 and T2 (feedback epoch), as any results after T2 could overlap with the next run temporally. However, feedback durations are different among different conditions, with longest for wins, shortest for LDWs, no feedback for losses. First, it is unclear the duration “range” of the feedback in these 4 conditions, also it is not clear about bonus feedback. So, it is better to clearly describe this information in Table (duration range). Furthermore, the main comparison is between the win/LDW/bonus and loss conditions. If there is no audiovisual feedback in the loss condition, the T1-T2 duration for the loss condition is likely overlapping with the next run temporally, how to perform this comparison properly? Thus, if the win and loss (or LDW) conditions have a relatively similar trial features (e.g., duration), it may be better to compare these two conditions only.

RESPONSE: We thank the reviewer for their careful consideration of this analysis. We agree that the different degrees of overlap between epochs referencing the start (T1) and end (T2) of audiovisual feedback limit our ability to discern event-related changes in pupil diameter. This is particularly the case for losses-disguised-as-wins. We have included the ranges for feedback duration in S1 Table under the column “Duration of Audiovisuals”. We have also indicated in the manuscript that, since loss trials do not have audiovisual feedback, there is no T2 event for this trial type present in the analyses. The fixed-effects regression approach we employed was partly chosen for its suitability to cases where some outcome types may not occur, or may not occur for all participants (Allison, 2012).

2. The authors conducted some other experiments to control luminance influence on pupil size, which is appropriate. However, there are many other visual factors that also influence pupil size (e.g., Barbur, J. L. (2004). Learning from the pupil-studies of basic mechanisms and clinical applications. The visual neurosciences, 1, 641-656). Besides, pupil dilation is observed after acoustic sounds, and the magnitude of dilation is scaled with stimulus intensity (also arousing level). All these factors need to be taken into consideration to make sure that the observed differences in pupil size are not simply mediated by these differences.

RESPONSE: Thank you for these excellent points. The reviewer is correct that pupil diameter may be impacted by other factors besides luminance and slot machine outcomes. Unfortunately, we do not have access to factors such as acoustic data in these 2 studies. Disentangling these factors is beyond the scope of the present study and would be best reserved for future directions. As such, we acknowledge the reviewer’s concerns as limitations in the paper (page 24): 

“Nevertheless, pupil diameter can be influenced by several factors. For example, the duration of the reel spin was substantially longer during free spin bonus features than other outcomes (see Fig 4 and S1 Table). Because pupil diameter is sensitive to uncertainty and anticipation of reward (Nassar et al., 2012; Pietrock et al., 2019), it is unclear from our findings whether the effects of bonus features are due to reward per se, or due to the anticipation of reward. We also note that the symbols that trigger bonus rounds on ‘Buffalo Spirit’ are physically larger and accompanied by loud and distinctive auditory flourishes. Recent research has shown that pupil dilation can be elicited by auditory and/or visual stimuli, and further modulated by stimulus salience (Wang & Munoz, 2015). Thus, the observed effects of bonus rounds on pupil size may be due to their auditory and visual features, as well as their relative salience and infrequency.” 

3. Also, trial number also radically different among different conditions, as most trials probably are loss trials (~60-70 %). Does trial number could affect the observed effects in pupil size?

RESPONSE: This is a good observation. As shown in S1 Table under the column “Total Events”, the number of trials does vary substantially across outcome types: Loss (n = 7451), Win (n = 876), losses-disguised-as-wins (n = 1016), and Bonus (n = 58). This discrepancy is because we cannot control the frequency of different outcomes and an unavoidable limitation of using a genuine slot machine in the experiment. We acknowledge this limitation and the potential consequences of infrequent bonuses and frequent losses: 

“In contrast to the bonuses, loss outcomes occurred on around 80% of trials, and this relative predictability is less than ideal for using the losses as the reference category for all three types of reinforcing outcomes. The low frequency of bonus rounds also means that these events occurred in only a subset of sessions and participants. Because we used a genuine slot machine, we could not control these outcome frequencies, and our analyses involving bonus features draw upon a very small number of observations (see S1 Table). This likely diminished the reliability of the reported coefficient estimates.” (Page 24)

Response to the Academic Editor

RESPONSE: Thank you for pointing this out. Efforts have been made to ensure that the manuscript is meeting PLOS ONE’s style requirements. 

(I have read the journal's policy and the authors of this manuscript have the following competing interests: 

The Centre for Gambling Research at UBC receives funding from the Province of British Columbia and the British Columbia Lottery Corporation (BCLC), a Canadian Crown Corporation. The slot machines used in the present study were provided to the Centre for Gambling Research by the BCLC. 

EHLO has received a speaker honorarium from the Massachusetts Council on Compulsive Gambling (USA). She has accepted travel or accommodation for speaking engagements from the National Council for Responsible Gambling (USA), the International Multidisciplinary Symposium on Gambling Addiction (Switzerland), and the Responsible Gambling Council (Canada). She has not received any further direct or indirect payments from the gambling industry or groups substantially funded by gambling.

MAF has received a speaker honorarium from the British Columbia Lottery Corporation (BCLC). 

MVC has received a speaker honorarium from the Responsible Gaming Association of New Mexico (USA).

LC is the Director of the Centre for Gambling Research at UBC. LC has received speaker/travel honoraria from the National Association of Gambling Studies (Australia) and the International Center for Responsible Gaming (USA). He has received academic consulting fees from Gambling Research Exchange Ontario (Canada), GambleAware (UK), and the International Center for Responsible Gaming (USA). He has not received any further direct or indirect payments from the gambling industry or groups substantially funded by gambling. He has received royalties from Cambridge Cognition Ltd. relating to neurocognitive testing.

AJK, WSM, KIM, JF and MS report no conflicts of interest.)

RESPONSE: As requested, we have included the following statement in our Competing Interests section: “This does not alter our adherence to PLOS ONE policies on sharing data and materials” (see Cover Letter).

3. We note that you have referenced) (Ferrari & Clark, unpublished) which has currently not yet been accepted for publication. Please remove this from your References and amend this to state in the body of your manuscript: (ie “Bewick et al. [Unpublished]”) as detailed online in our guide for authors

RESPONSE: As suggested, (Ferrari & Clark. [Unpublished]) is in the body of our manuscript and excluded from the References.

4. We note that Figure 3 in your submission contain copyrighted images. All PLOS content is published under the Creative Commons Attribution License (CC BY 4.0), which means that the manuscript, images, and Supporting Information files will be freely available online, and any third party is permitted to access, download, copy, distribute, and use these materials in any way, even commercially, with proper attribution. For more information, see our copyright guidelines: http://journals.plos.org/plosone/s/licenses-and-copyright.

RESPONSE: The copyright holders (WMS / Scientific Games) declined our copyright request. As such, we have supplied a replacement figure that we created (see below, Fig 3). We confirm that Fig 3 is entirely created by the authors and does not contain any copyrighted material, illustrations, designs, etc.

Fig 3. Illustrations depicting the sequence of on-screen events on free-spin bonus features and winning outcomes, for the two slot machine models used in the present study. These illustrations capture the approximate colour schemes and image dimensions of the key game elements. Researchers requiring actual screenshots should contact the corresponding author. Losses-disguised-as-wins follow similar in-game processes to the winning outcomes shown, and loss outcomes are similar to wins during the reel spin but do not contain any audiovisual feedback. Note the contrasting luminance properties for Buffalo Spirit (as a light screen background) and Ice Empress (with a darker screen background). 

RESPONSE: Thank you. Captions are now included for the Supporting Information files at the end of our manuscript. In-text citations are updated accordingly.

---

## [Decision Letter · Decision Letter 1]

13 Jul 2022

Do pupillary responses during authentic slot machine use reflect arousal or screen luminance fluctuations? A proof-of-concept study

PONE-D-22-00740R1

Dear Dr. Kim,

We’re pleased to inform you that your manuscript has been judged scientifically suitable for publication and will be formally accepted for publication once it meets all outstanding technical requirements.

Kind regards,

Manuel Spitschan

Academic Editor

PLOS ONE

Additional Editor Comments (optional):

Reviewers' comments:

Reviewer's Responses to Questions

**Comments to the Author**

1. If the authors have adequately addressed your comments raised in a previous round of review and you feel that this manuscript is now acceptable for publication, you may indicate that here to bypass the “Comments to the Author” section, enter your conflict of interest statement in the “Confidential to Editor” section, and submit your "Accept" recommendation.

Reviewer #1: All comments have been addressed

Reviewer #2: All comments have been addressed

2. Is the manuscript technically sound, and do the data support the conclusions?

Reviewer #1: Yes

Reviewer #2: Partly

3. Has the statistical analysis been performed appropriately and rigorously? 

Reviewer #1: Yes

Reviewer #2: Yes

4. Have the authors made all data underlying the findings in their manuscript fully available?

Reviewer #1: Yes

Reviewer #2: No

5. Is the manuscript presented in an intelligible fashion and written in standard English?

Reviewer #1: Yes

Reviewer #2: Yes

6. Review Comments to the Author

Reviewer #1: I appreaciate the work that has been done to address the issues raised, in particular the supplementary info on pupil forshortening corrections. I have no furter comments or suggestions.

Reviewer #2: The authors have addressed all my previous concerns, but as mentioned, many factors known to change pupil size are different among different conditions (e.g., visual and auditory attributes, trial number, trial duration). I'm less certain about their conclusion, nevertheless, it is indeed needed to see more pupil results in the real world data in gambling.

7. PLOS authors have the option to publish the peer review history of their article (what does this mean?). If published, this will include your full peer review and any attached files.

Reviewer #1: **Yes: **Joel T. Martin

Reviewer #2: No

---

## [Editor Report · Acceptance letter]

15 Jul 2022

PONE-D-22-00740R1 

Do pupillary responses during authentic slot machine use reflect arousal or screen luminance fluctuations? A proof-of-concept study 

Dear Dr. Kim:

I'm pleased to inform you that your manuscript has been deemed suitable for publication in PLOS ONE. Congratulations! Your manuscript is now with our production department. 

Kind regards, 

on behalf of

Dr. Manuel Spitschan 

Academic Editor

PLOS ONE